# Near-Infrared Photoimmunotherapy (NIR-PIT) in Urologic Cancers

**DOI:** 10.3390/cancers14122996

**Published:** 2022-06-17

**Authors:** Hiroshi Fukushima, Baris Turkbey, Peter A. Pinto, Aki Furusawa, Peter L. Choyke, Hisataka Kobayashi

**Affiliations:** 1Molecular Imaging Branch, Center for Cancer Research, National Cancer Institute (NIH), Bethesda, MD 20892, USA; hiroshi.fukushima@nih.gov (H.F.); ismail.turkbey@nih.gov (B.T.); aki.furusawa@nih.gov (A.F.); pchoyke@mail.nih.gov (P.L.C.); 2Urologic Oncology Branch, Center for Cancer Research, National Cancer Institute (NIH), Bethesda, MD 20892, USA; pintop@mail.nih.gov

**Keywords:** near infrared photoimmunotherapy, urologic cancers, endoscopy, target protein

## Abstract

**Simple Summary:**

Near-infrared photoimmunotherapy (NIR-PIT) is a novel molecularly-targeted therapy that induces rapid cancer cell death by systemically administering an antibody-photoabsorber conjugate (APC) that binds to cancer cells and irradiating NIR light that drives photochemical transformations of the APC. APCs are constructed by using a monoclonal antibody targeting a cancer cell surface antigen and conjugating it to IRDye700DX silica-phthalocyanine dye. NIR-PIT can selectively kill cancer cells while leaving normal tissues unaffected. Moreover, NIR-PIT activates anti-cancer immunity through the induction of immunogenic cell death of cancer cells. Currently, NIR-PIT is being applied clinically in head and neck squamous cell carcinoma. In previous preclinical studies, NIR-PIT showed excellent efficacy against urologic cancers including bladder cancer and prostate cancer. The clinical application of NIR-PIT will expand to urologic cancers in the near future.

**Abstract:**

Near-infrared photoimmunotherapy (NIR-PIT) is a novel molecularly-targeted therapy that selectively kills cancer cells by systemically injecting an antibody-photoabsorber conjugate (APC) that binds to cancer cells, followed by the application of NIR light that drives photochemical transformations of the APC. APCs are synthesized by selecting a monoclonal antibody that binds to a receptor on a cancer cell and conjugating it to IRDye700DX silica-phthalocyanine dye. Approximately 24 h after APC administration, NIR light is delivered to the tumor, resulting in nearly-immediate necrotic cell death of cancer cells while causing no harm to normal tissues. In addition, NIR-PIT induces a strong immunologic effect, activating anti-cancer immunity that can be further boosted when combined with either immune checkpoint inhibitors or immune suppressive cell-targeted (e.g., regulatory T cells) NIR-PIT. Currently, a global phase III study of NIR-PIT in recurrent head and neck squamous cell carcinoma is ongoing. The first APC and NIR laser systems were approved for clinical use in September 2020 in Japan. In the near future, the clinical applications of NIR-PIT will expand to other cancers, including urologic cancers. In this review, we provide an overview of NIR-PIT and its possible applications in urologic cancers.

## 1. Introduction

Near-infrared photoimmunotherapy (NIR-PIT) is a new molecularly-targeted anti-cancer therapy that uses NIR light to induce photochemical reactions within antibody-photoabsorber conjugates (APCs), thereby leading to rapid cell death [1,2]. To synthesize APCs, the photoabsorber IRDye700DX (IR700), a silica-phthalocyanine dye, is conjugated to a monoclonal antibody that targets cancer-specific proteins on the cell surface [3]. Approximately one day after intravenous infusion of an IR700-based APC, NIR light is delivered to the cancer, activating a photochemical reaction that results in highly selective cancer cell killing while leaving normal tissues unaffected [4,5,6,7]. NIR-PIT has been used to target various cancer-specific proteins, such as epidermal growth factor receptor (EGFR) [8,9,10,11,12,13] and human epidermal growth factor receptor-2 (HER2) [14,15,16,17,18,19], and both have shown excellent anti-cancer effects. Furthermore, the clinical use of EGFR-targeted NIR-PIT progressed into clinical trials of patients with head and neck squamous cell carcinoma (HNSCC). A phase I/IIa, first-in-human, open-label, multicenter study of EGFR-targeted NIR-PIT using a cetuximab-IR700 conjugate (RM-1929) was completed with tolerable and manageable side effects in patients with heavily pretreated recurrent HNSCC [20]. A global phase III randomized controlled trial of EGFR-targeted NIR-PIT for recurrent HNSCC was started in 2018 and is currently ongoing (ClinicalTrials.gov Identifier: NCT03769506). In Japan, the first APC for human use, a cetuximab-IR700 conjugate (AkaluxTM; Rakuten Medical Inc., San Diego, CA, USA) and a NIR laser system (BioBlade^TM^; Rakuten Medical Inc., San Diego, CA, USA) were approved for clinical use by the Pharmaceuticals and Medical Devices Agency (PMDA) in September 2020. In the US, a clinical trial of EGFR-targeted NIR-PIT in newly diagnosed HNSCCs and SCC of the skin recently got underway. These major advances presage the application of NIR-PIT to other cancers, including urologic cancers. In this review, we present an overview of NIR-PIT and discuss how it could be applied to treat urologic cancers in the clinic based on extensive preclinical data. We also highlight potential target molecules for NIR-PIT in urologic cancers.

## 2. NIR-PIT

In 1983, the term “photoimmunotherapy” was used to imply a targeted photodynamic therapy using antibodies that kill target cells based on reactive oxygen species (ROS)-induced cytotoxicity using antibodies that are conjugated to conventional photosensitizers such as hematoporphyrin [21]. However, it was not successfully used as an anti-cancer treatment via systemic administration [22]. This is because the hydrophobic nature of conventional photosensitizers causes rapid liver accumulation of conjugates, leading to insufficient delivery of conjugates to the tumor [23]. Moreover, because the cell killing efficacy of this “old photoimmunotherapy” depends on ROS that are mostly produced out of the target cells, it causes significant non-specific damage to adjacent normal cells. In contrast, NIR-PIT selectively kills APC-bound target cells by the immunogenic cell death as a result of photo-induced ligand release reactions rather than ROS reactions after an intravenous injection of IR700-based APCs (Figure 1). As a consequence, an anti-cancer host–immune response is rationally induced after NIR-PIT. Thus, NIR-PIT totally differs from the “old photoimmunotherapy”. Compared to conventional anti-cancer therapies such as surgery, radiotherapy, and chemotherapy, NIR-PIT has several therapeutic advantages, principally its highly selective nature and its enhancement of anti-cancer immunity. In this section, we summarize what is known regarding the mechanism of NIR-PIT.

### 2.1. Mechanism of NIR-PIT

NIR-PIT employs IR700-based APCs, which are antibodies conjugated to the IR700 silica-phthalocyanine dye and absorb NIR photons [3]. When injected intravenously, IR700-based APCs bind to a specific cell membrane antigen on a cancer cell’s surface. This alone is insufficient as a treatment effect. However, approximately 24 h after injection, the tumor is exposed to 690 nm NIR light from a laser, usually in doses of 30–100 J/cm^2^ [4,5]. NIR light induces the axial ligands of the IR700 molecule to disengage from the molecule, changing the entire APC from a very hydrophilic to a very hydrophobic molecule (Figure 1) [7,24]. This alteration in IR700’s chemical properties encourages APC aggregation and conformational changes that damage the cell membrane [7,24,25]. Even while the light is still being delivered, cancer cells can be observed to swell, bleb, and burst, releasing their contents into the extracellular environment and causing necrotic cell death [26]. This cytotoxic mechanism distinguishes NIR-PIT from conventional photodynamic therapy (PDT) and photothermal therapy (PTT), which require the generation of ROS to cause non-specific damage to adjacent normal tissues [27,28]. The highly selective cell killing of NIR-PIT allows it to be repeated for residual or recurrent tumors without damaging normal adjacent tissue [29].

The effectiveness of NIR light activation is limited by the short distance it can penetrate into tissue. NIR light can penetrate up to about 2 cm below the tissue surface but is optimal within 1 cm, as light attenuation is significant beyond that point. This indicates that tumors near the body or mucosal surface are suited to NIR-PIT using an externally applied frontal light diffuser [30]. Although NIR light can be transmitted far further through the air in the lungs in the case of the treatment of lung and pleural cancers [15,31], it is rapidly attenuated in most solid tissues, hence the light source must be positioned either close to or within tumors using interstitial optical fibers [32]. Thus, tumors deep within the body are difficult to treat using a frontal light diffuser alone. A cylindrical light diffuser-fiber inserted into the treatment site can overcome this limitation by bringing light to the center of the tumor [10,14,33,34]. Using a cylindrical light diffuser, NIR light can be delivered in a cylindrical pattern (radially 1 cm from the probe for a total distance of 2 cm) to practically any tumor site via a needle, catheter, or endoscope.

### 2.2. Activation of Anti-Cancer Immunity

NIR-PIT enhances anti-cancer immunity through the induction of immunogenic cell death (ICD) [35,36]. ICD is a specific type of cell death in which adaptive immune cells react to a barrage of cancer-specific antigens released from injured cancer cells [37,38,39]. Most other cancer therapies, such as chemotherapy and radiotherapy, cause apoptotic cell death [40,41], which is generally more organized or “programmed” and less immunogenic [42,43]. In ICD, injured cancer cells, in addition to releasing tumor antigens, release various danger signals including high mobility group box 1 (HMGB1), calreticulin (CRT), heat shock protein (HSP) 90, HSP 70, and adenosine triphosphate (ATP) [37,38,39]. These danger signals cause immature dendritic cells (DCs) to begin presenting cancer-specific antigens to T cells, thus becoming mature DCs [35]. These activated DCs prime and educate naïve T cells to become cancer-specific cytotoxic T cells [44]. In contrast to apoptosis, NIR-PIT-induced ICD strongly activates the host immune system that works in concert with the direct killing by NIR-PIT to selectively eradicate cancer cells [35]. Thus, NIR-PIT can efficiently provoke ICD and eventually activate anti-cancer immune cells that may have systemic effects as well. This can result in abscopal effects in sites of disease, in addition to the treatment field. Furthermore, NIR-PIT can transform a poorly immunogenic tumor into a highly immunogenic tumor by enhancing anti-cancer immunity [45]. Therefore, it is rational that when NIR-PIT is combined with immune activation therapies such as immune checkpoint inhibitors (ICIs), it shows synergistic efficacy and even abscopal effects in immunocompetent mice [44,45,46,47].

### 2.3. Selective Depletion of Immune Suppressive Cells or Cancer-Associated Fibroblasts

In addition to targeting cancer cells, NIR-PIT can selectively target cells in the tumor microenvironment, including immune cells and cancer-associated fibroblasts. This can be advantageous when seeking to amplify the NIR-PIT-induced immune response. The selective depletion of immune suppressive cells can be achieved by targeting surface antigens that are unique to each cell type. For instance, regulatory T cells (Tregs) can be spatiotemporally depleted by NIR-PIT targeting CD25 or cytotoxic T-lymphocyte antigen 4 (CTLA4), both of which are overexpressed on the surface of Tregs [48,49,50,51]. Fibroblast activation protein (FAP)-targeted NIR-PIT provides selective depletion of cancer-associated fibroblasts, which promote cancer growth and facilitate drug resistance [52,53]. These types of non-tumor-targeted NIR-PIT can have significant anti-cancer effects in syngeneic mouse models of cancer where the immune system is intact.

### 2.4. Super-Enhanced Permeability and Retention (SUPR) Effects

Immediately after NIR-PIT, the vascular permeability in the tumor bed is significantly increased, especially for nano-sized molecules. It is well known that most tumors have relatively leaky vessels and this phenomenon is called the “enhanced permeability and retention” (EPR) effect [54,55]. This effect is relatively modest, representing less than 5% difference compared with normal tissue. On the other hand, after NIR-PIT, permeability can reach an astounding 24-fold increase compared with normal tissue, especially for nanoparticles. This effect has been dubbed the “super-enhanced permeability and retention” (SUPR) effect [56,57,58,59]. The SUPR effect is triggered by the death of perivascular cancer cells, which are the first to be exposed to APCs after the intravenous infusion of APCs, and thus is the first to be killed after the administration of NIR light. A gap between the vessel wall and the residual tumor mass is created as a result of this killing, leading to vessel enlargement, increased blood volume, decreased blood velocity, and the reduction of vascular resistance, thereby enabling nano-sized compounds to be more efficiently transported into the tumor bed, where they can remain for several days. If the nano-sized compound is a therapeutic agent, it can be used to treat cancer cells not killed by the initial NIR-PIT, but at a much lower dose than would be required without SUPR. For example, when NIR-PIT is combined with nano-sized agents such as liposome-containing daunorubicine or albumin-bound paclitaxel, therapeutic effects were significantly augmented compared with either therapy alone [56,60]. Therefore, combining NIR-PIT with nano-sized anti-cancer agents is more effective than using either therapy alone because the increased leakiness of the vessels allows for higher drug concentrations and longer exposure of the residual tumor to the drug.

## 3. NIR-PIT for Urologic Cancers

In theory, all urologic cancers can be treated with NIR-PIT by selecting optimal APCs and delivering NIR light to the treatment site using a cylindrical light diffuser. In some ways, urologic tumors are ideally suited to NIR-PIT as many of the urologic cancers are approachable either by endoscopy, needles, or laparoscopy. Thus, light delivery, which might be limiting in other cancers, is not a significant barrier in urology. In this section, we discuss the potential applications of NIR-PIT in urologic cancers (Table 1).

### 3.1. Bladder Cancer

Bladder cancer is the tenth most common cancer worldwide, with 573,278 new cases and 212,536 deaths in 2020 [62]. Bladder cancer accounts for over 90% of urothelial cancer, and its most common histology is urothelial carcinoma.

#### 3.1.1. Application of NIR-PIT to Bladder Cancer

Bladder tumors are diagnosed with cystoscopy and initially treated with transurethral resection of the bladder tumor (TURB) via a cystoscope [63]. In addition, cystoscopy is also useful in the detection of recurrent tumors after TURB. Various strategies are used in the setting of recurrent disease, but it is quite common, and thus requires frequent monitoring by cystoscopy.

Given that bladder cancers are directly approachable via cystoscopy, NIR-PIT via cystoscopy could be an advantageous treatment modality (Figure 2A). NIR light could be introduced via cystoscopy and distributed equally around the inner surface of the bladder using an appropriate diffuser at the tip of the fiber optic probe. Since NIR light can penetrate up to 2 cm from the tissue surface [30], NIR-PIT can eradicate not only surface small tumors and carcinoma in situ (CIS) lesions, but also deeper cancer cells that become targetable after resection of the bulk of the tumor.

Bladder cancer is categorized by the presence of muscle invasion of the bladder wall. The two types of bladder cancer are non-muscle-invasive bladder cancer (NMIBC) and muscle-invasive bladder cancer (MIBC). NMIBC is not a lethal disease, and it is generally treated with bladder-preservation therapy, including TURB [63]. Nevertheless, NMIBC has high recurrence rates ranging from 38% to 65% [64,65,66], and thus TURB needs to be repeated many times, which may cause perforation and extensive scarring of the bladder [67]. Moreover, high-risk NMIBC, including CIS, is hard to cure with TURB alone, and thus a form of immunotherapy is used: intravesical instillation of Bacille Calmette-Guérin (BCG) [63]. BCG therapy is thought to induce a host immune response that controls recurrence. Although BCG therapy occasionally induces severe adverse events, most patients tolerate it well; however, approximately 50% of patients with high-risk NMIBC recur and 10% eventually progress to MIBC [68,69]. Therefore, novel therapeutic options for NMIBC are an unmet need. Recently, fluorescence has been used to aid the diagnosis during cystoscopy. An oral photosensitizer such as 5-aminolevulinic acid and hexaminolevulinate is administered and, during cystoscopy, blue light irradiation is used to increase the detection of sites of recurrence [63]. In theory, this agent can also be used for photodynamic therapy (PDT). However, PDT has not been widely accepted for bladder cancer treatment due to non-selective damage and high rates of severe adverse events [70]. Unlike PDT, NIR-PIT selectively kills cancer cells without damaging non-target cells (e.g., normal urothelium in the bladder), and thus NIR-PIT could be adopted for bladder cancer treatment.

MIBC is a more aggressive disease than NMIBC, and the gold standard for MIBC is neoadjuvant chemotherapy followed by radical cystectomy [71]. However, its prognosis is unfavorable with a cancer-specific survival probability at 5 years of less than 60% [72]. NIR-PIT could be used in the neoadjuvant setting to improve postoperative oncological outcomes. Moreover, radical cystectomy has considerable morbidity, with reported complication rates ranging from 32% to 54% [73,74]. Recently, multimodal bladder-preservation therapy has been acknowledged as a less invasive but similarly effective treatment option for MIBC [75,76]. Multimodal bladder-preservation therapy generally includes maximal TURB and chemoradiotherapy [77]. Given its highly selective cytotoxicity towards cancer cells, NIR-PIT alone or in combination with other therapies may play a pivotal role in bladder-preservation therapy against MIBC. NIR-PIT could also be applied in metastatic settings to treat or palliate disease. ICD induced by NIR-PIT activates anti-cancer immunity that could be further enhanced in combination with immune activation therapies, including ICIs [44,45,46,47]. Therefore, NIR-PIT in combination with ICIs may be an effective way of treating metastatic bladder cancer.

#### 3.1.2. Target Molecules for NIR-PIT in Bladder Cancer

##### EGFR

EGFR is one member of the erythroblastosis oncogene B (erbB) family of tyrosine kinase receptors [78]. The physiological role of EGFR includes promoting epithelial tissue growth and homeostasis. EGFR mutations and/or overexpression have been found in a variety of human malignancies, and EGFR-targeted therapy has become a standard of care in several cancers [79]. EGFR is the most promising target molecule since it is overexpressed in 55–74% of bladder cancer tissues [80,81,82]. EGFR-targeted NIR-PIT provokes cancer cell death in EGFR-expressing human bladder cancer cells in vitro and suppresses tumor growth in mice xenografts from the same cell lines [83]. EGFR-targeted NIR-PIT was also effective for EGFR-expressing canine bladder cancer cells both in vitro and in vivo [84]. EGFR-targeted NIR-PIT has been used successfully in humans with HNSCCs and is in phase III studies worldwide. Therefore, this existing NIR-PIT APC could be applied to bladder cancer as well.

##### HER2

HER2, which is also known as ERBB2, HER2/neu, or c-erbB2, is another member of the ErbB family of tyrosine kinase receptors [78]. HER2 overexpression is observed in 38–53% of bladder cancers [80,82,85,86]. A combination of EGFR- and HER2-targeted NIR-PIT had a higher efficacy than either type of NIR-PIT in bladder cancer xenografts alone [87]. It is unlikely that HER2-targeted NIR-PIT alone will be successful in bladder cancer, but it could be used as part of a “cocktail” of APCs injected prior to NIR light administration in the bladder.

##### CD44

CD44 is a glycosylated membrane receptor that is involved in cellular adhesion and signaling [88]. CD44 is known as one of the cancer stem cell markers and is associated with resistance to various anti-cancer therapies [88]. For instance, CD44 is implicated in the development of cisplatin and radiation resistance in bladder cancer [89,90]. CD44 is highly expressed in 52% of bladder cancers [91]. Thus, CD44-targeted NIR-PIT presents the opportunity to eliminate cancer cells, including cancer stem cells, in particular, thereby reducing the likelihood of recurrence. A preclinical study showed that CD44-targeted NIR-PIT significantly inhibited tumor development and improved survival in syngeneic mouse models of oral squamous cell carcinoma [92]. Furthermore, in syngeneic mouse models, combinations of CD44-targeted NIR-PIT and ICIs proved more effective than either therapy alone [44,45,93].

##### CD47

CD47 is a cell surface protein that promotes the migration of neutrophils and T cells and prevents macrophages from phagocytizing cancer cells (so-called “don’t eat me” signaling) [94]. CD47 expression is elevated in approximately 70% of bladder cancers, but is absent in normal urothelium [95]. CD47-targeted NIR-PIT showed efficacy against human bladder cancer cell lines and patient-derived bladder cancer cells in vitro and in xenograft mouse models [96]. Thus, CD47 is a promising target for bladder NIR-PIT.

##### Tumor-Associated Calcium Signal Transducer 2 (TROP-2)

TROP-2, a glycoprotein involved in intracellular calcium signal transduction [97], is overexpressed in approximately 80% of urothelial carcinomas [98]. The efficacy of TROP-2-targeted NIR-PIT has already been shown in human pancreatic carcinoma and cholangiocarcinoma cells in vivo [99], but not bladder cancer. Given that a phase II clinical trial showed an encouraging efficacy of sacituzumab govitecan, a TROP-2-targeted antibody-drug conjugate, in the treatment of advanced bladder cancer [100], TROP-2 may be a good target molecule for NIR-PIT in bladder cancer.

##### Programmed Death-Ligand 1 (PD-L1)

The programmed cell death protein 1 (PD-1)/PD-L1 axis is an inhibitory signaling pathway that facilitates the immune evasion of cancer cells in the tumor microenvironment. PD-L1 expression is observed in various normal cells, including vascular endothelial cells, smooth muscle cells, hepatocytes, pancreatic islet cells, mesenchymal stem cells, and immune cells such as B cells, T cells, dendritic cells, and macrophages. In addition, PD-L1 is overexpressed in various cancers, including bladder cancer. PD-L1-targeted NIR-PIT using avelumab, which is a humanized monoclonal antibody against PD-L1, significantly suppressed tumor formation and prolonged survival in xenograft mouse models [101]. PD-L1-targeted NIR-PIT depleted PD-L1 expressing tumor-associated macrophages and cancer cells in an ovarian cancer mouse model [102]. Furthermore, PD-L1-targeted NIR-PIT eliminated cancer cells through the activation of anti-cancer immune reactions in syngeneic mouse models of cancer [103]. Therefore, PD-L1-targeted NIR-PIT is an attractive target in the treatment of bladder cancer.

##### CTLA4 or CD25

Given that immunotherapy, including BCG therapy, is effective in bladder cancer, Treg-targeted NIR-PIT may be a therapeutic option that also activates host immunity in bladder cancer. Tregs are expressed in the microenvironment of bladder cancer and are associated with poor therapeutic responses to BCG therapy [104]. Thus, CTLA4- or CD25-targeted NIR-PIT, which selectively depletes Tregs [48,49,50,51,93], may activate host immunity, thereby resulting in improved efficacy. Unlike tumor targeted NIR-PIT, this treatment will not work by direct cancer cell killing, but rather by inducing anti-cancer host immunity to kill cancer cells. As such, it holds promise as an immune regulator of the bladder wall, potentially improving bladder cancer surveillance.

##### Selection of Target Molecules for NIR-PIT According to Molecular Subtypes

Since bladder cancer is a molecularly heterogeneous disease characterized by genomic instability and high mutation rates [105], molecular subtype classification may be useful in selecting target molecules for NIR-PIT. Although many studies classify bladder cancer according to molecular subtypes based on transcriptome profiling [106,107], cell-surface proteomic profiling is the most important information in identifying potential target molecules for NIR-PIT. Thus, the Lund classification, a molecular subtype classification based on immunohistochemistry and tissue microarray, could be helpful among the classifications proposed so far [108,109]. In the Lund classification, bladder cancer is mainly categorized into three groups: (1) Genomically Unstable (GU), (2) Basal/Squamous cell carcinoma (SCC)-like, and (3) Urothelial-like (Uro). GU tumors have high expressions of CDKN2A and HER2. Thus, HER2-targeted NIR-PIT may be effective in this subgroup. In Basal/SCC-like tumors, which are characterized by high expressions of P-cadherin and DSC2/3, EGFR expression is highly expressed. Thus, EGFR-targeted NIR-PIT may be a good therapeutic option for this group. As for Uro tumors, which highly express FGFR3 and CCND1, EGFR expression is restricted to the basal cell layers of tumors, suggesting that FGFR3 might be a good target molecule for NIR-PIT in this group. To develop efficacious NIR-PIT regimens for bladder cancer, future studies should elucidate optimal combinations of target molecules for NIR-PIT according to molecular subtypes.

The promise of NIR-PIT of the bladder is that it could be a relatively non-invasive therapy and delivered cystoscopically, which could be repeated at regular intervals. The immune response, which can be potentially amplified by coupling a tumor-targeted APC with an immunosuppressive-targeted APC, could provide a hostile immune environment for the spread and recurrence of bladder cancer.

### 3.2. Upper Tract Urothelial Cancer

Upper tract urothelial cancer, including cancers of the renal pelvis and ureter, is particularly difficult to treat. Such tumors account for only 5–10% of all urothelial cancers [110]. The most common histologic type is urothelial carcinoma. The diagnosis of upper tract urothelial cancer is typically based on a combination of radiologic studies and urine cytology. Diagnostic ureterorenoscopy with biopsy has been increasingly utilized in the diagnosis. For this reason, NIR-PIT could be a viable treatment strategy.

#### 3.2.1. Application of NIR-PIT to Upper Tract Urothelial Cancer

Although the gold standard for treatment of patients with non-metastatic upper tract urothelial cancer is radical nephroureterectomy (RNU), this causes significant decreases in renal function, which may, in turn, affect the selection of adjuvant systemic therapy [111]. Thus, endoscopic resection or laser ablation is a viable option to spare renal function in carefully selected patients with low-grade and low-stage tumors [112,113]. However, a risk of disease progression remains with endoscopic management because of the suboptimal sensitivity of presurgical imaging and biopsy [114]. Given that NIR light deeply penetrates tissues, NIR-PIT using a cylindrical light diffuser introduced via a ureterorenoscope may be a practical way to achieve the complete eradication of cancer cells and improve oncological outcomes of endoscopic treatment. Furthermore, NIR-PIT may be clinically applicable to more advanced diseases. NIR-PIT may be used as neoadjuvant therapy for locally advanced disease prior to surgery to debulk the tumor. In metastatic settings, NIR-PIT may be combined with immune activation therapies such as ICIs.

#### 3.2.2. Target Molecules for NIR-PIT in Upper Tract Urothelial Cancer

Upper tract urothelial cancer has a similar histology to bladder cancer, and thus its molecular features are partially shared. Hence, target molecules for bladder cancer would likely be applicable to upper tract urothelial cancers. High EGFR and HER2 expression were observed in 43–55% and 35–37% of the surgical specimens of upper tract urothelial cancer [115,116,117,118,119]. Thus, as in bladder cancer, EGFR- and HER2-targeted NIR-PIT would be good target molecules for NIR-PIT in upper tract urothelial cancer.

### 3.3. Prostate Cancer

Prostate cancer is the second most common cancer in males, with 1,414,259 new cases worldwide, and it is the fifth most common cause of cancer death, with 375,304 deaths worldwide in 2020 [62]. The mortality rate of prostate cancer is discordant with its high incidence rate in part due to the substantial number of prostate cancers detected by population-based prostate-specific antigen (PSA) screening, and in part due to the high prevalence of indolent, slow growing cancers [120]. Nonetheless, prostate cancer is a major cause of death and new therapies are needed.

#### 3.3.1. Application of NIR-PIT to Prostate Cancer

Whole-gland treatments, including radical prostatectomy and radiotherapy, are currently utilized to definitively treat organ-confined prostate cancer. However, there is growing evidence that many low- and intermediate-risk prostate cancer patients are overtreated, resulting in side effects from therapy without deriving benefit [121]. Radical prostatectomy and radiotherapy are associated with significant morbidities, including urinary, sexual, and bowel dysfunction [122]. Active surveillance has been widely accepted as a standard of care to decrease the risk of unnecessary definitive treatments for patients with low-risk prostate cancer, but approximately 20–30% of men discontinue active surveillance and select definitive treatment mainly due to disease progression during surveillance [123]. Thus, there is an unmet need for an alternative to whole-gland treatments, especially in patients with low-to-intermediate risk prostate cancer. Image guided focal therapy using a variety of ablative technologies, including high-intensity focal ultrasound, laser ablation, irreversible electroporation, cryotherapy etc., have emerged as an alternative to whole gland therapy for prostate cancers [124]. These focal therapies commonly utilize magnetic resonance imaging (MRI) for guidance of the ablation, but MRI is well known to underestimate the extent of disease burden [125]. The literature suggests that focal therapy approaches using MRI guidance can result in treatment failure due to incomplete ablation of the tumor [126,127]. Additionally, the application of focal therapy may not be possible for some tumors located close to the urethra, rectum, or external sphincter since these critical structures may be damaged during ablative therapies.

NIR-PIT is a promising therapeutic option for organ-confined prostate cancer. NIR-PIT can selectively eradicate cancer cells with minimal damage to functionally important tissues such as the periprostatic nerve network and the urethra. Nevertheless, NIR-PIT can treat all lesions in the prostate by delivering NIR light interstitially using a cylindrical light diffuser inserted via needles, which are uniformly placed in the prostate or directed into the region of abnormality seen in imaging studies (Figure 2B). A significant advantage of NIR-PIT over conventional focal therapy or partial gland ablation for organ-confined prostate cancer is its selectivity, which means that NIR light can be administered non-specifically within the prostate but will only be effective where APCs have bound to tumors [128,129]. Thus, NIR-PIT of prostate gland will be significantly more selective than existing focal therapy techniques that depend on tissue ablation. Like other therapeutic modalities, multiparametric MRI and MRI-guided or MRI-ultrasound fusion prostate biopsy can be helpful in therapy planning. MRI can define the presence of disease, and MRI-guided or MRI-ultrasound fusion prostate biopsy can contribute to prostate cancer diagnosis and accurate biopsy grading [130]. In one scenario, NIR-PIT of the prostate can be guided by prostate-specific membrane antigen (PSMA) positron emission tomography (PET), which depicts the site of more aggressive disease. One advantage of PSMA-PET is that it can also be used to monitor the success of therapy as PSMA-targeted NIR-PIT would be anticipated to significantly reduce PSMA-PET uptake.

NIR-PIT may be utilized to treat local recurrence following definitive local therapy, including surgery or radiotherapy. Moreover, NIR-PIT may have a role in the treatment of advanced prostate cancer. For locally advanced disease, NIR-PIT, as a neoadjuvant therapy, can shrink the tumor, which may then enable complete resection of the prostate by radical prostatectomy. For metastatic prostate cancer, NIR-PIT may have a role in multidisciplinary treatment strategies. ICIs provided suboptimal objective response rates (3–17%) in prostate cancer [131]. This is partially because prostate cancer has a low mutational burden and is recognized as a poorly immunogenic tumor with a minimal infiltration of T cells [131]. Since NIR-PIT can transform poorly immunogenic tumors into highly immunogenic tumors [45], NIR-PIT in combination with ICIs may be a theoretically sound strategy in the management of metastatic prostate cancer [44,45,46,47].

#### 3.3.2. Target Molecules for NIR-PIT in Prostate Cancer

##### PSMA

PSMA is a type 2 integral membrane glycoprotein that is highly expressed in the majority of prostate cancer cells [132]. PSMA expression is low in normal prostate tissues and in low grade tumors, but is also low in prostate cancers with neuroendocrine differentiation. However, in the majority of prostate cancers, PSMA expression is high, especially in metastatic or castration-resistant prostate cancer [133]. Thus, PSMA is an ideal ligand for radionuclide imaging and drug delivery strategies [132,134]. PSMA-targeted NIR-PIT significantly suppressed tumor growth and prolonged survival in human prostate cancer xenograft models [135]. PSMA-targeted NIR-PIT, using an anti-PSMA diabody or anti-PSMA minibody, showed a comparable therapeutic efficacy to antibody-based NIR-PIT both in vitro and in vivo [136]. Moreover, PSMA-targeted NIR-PIT using low-molecular-weight ligands showed cytotoxic effects on human prostate cancer cells in vitro [137].

The availability of PSMA-PET scans presents an interesting opportunity to guide and assess PSMA-targeted NIR-PIT (Figure 3) [138]. PSMA-PET scans can accurately identify the location of the tumor based on PSMA expression [139,140]. Additionally, PSMA-PET is specific for clinically significant prostate cancers with almost no false positives [139,141]. Thus, PSMA-PET could be used to select candidates for PSMA-targeted NIR-PIT. PSMA-PET-based image guidance could allow for precise transperineal placement of cylindrical light diffusers within the prostate. Moreover, therapeutic responses to PSMA-targeted NIR-PIT could be evaluated by PSMA-PET images and clinicians could determine whether additional NIR-PIT may be necessary or not.

##### Other Target Molecules

Although CD44 expression is rare in adenocarcinomas of the prostate, it is highly expressed in all cases of neuroendocrine prostate cancer (NEPC) [142]. Thus, CD44-targeted NIR-PIT is considered a good therapeutic option for NEPC, at least when it is localized within the pelvis. Since treatment-emergent NEPC can occur as a result of androgen deprivation therapy [143], CD44-targeted NIR-PIT may play a role as a local therapy for non-metastatic or oligo-metastatic castration-resistant prostate cancer with neuroendocrine differentiation.

Glypican-1 (GPC1) is a heparan sulfate proteoglycan that is highly expressed in human prostate cancer cell lines [144]. GPC1-targeted NIR-PIT significantly reduced the viability of GPC1-expressing human prostate cancer cells in vitro [145]. Thus, GPC1 may be a candidate target molecule for NIR-PIT in prostate cancer. It would be important to establish the degree of overlap between PSMA, GPC1, and CD44 to understand under what circumstances these agents could be combined to provide better coverage of the prostate tumor.

Although prostate cancer is considered an immunologically “cold” tumor, scattered Tregs and other T cells exist in the tumor microenvironment. Targeting such cells could activate the immune system, converting the cold environment to a more active immunologic environment. A preclinical study showed that CD25-targeted NIR-PIT depleted Tregs in murine prostate cancer cells, resulting in reduced bioluminescence in these luciferase-expressing cells in vivo [49]. Thus, CD25-targeted NIR-PIT may play a role in the treatment of prostate cancer as an adjuvant to tumor-targeted NIR-PIT.

### 3.4. Renal Cell Carcinoma

Kidney cancer is the ninth most common cancer worldwide. There are approximately 431,288 new cases and 179,368 deaths annually across the world [62]. The most predominant histology of kidney cancer is clear cell renal cell carcinoma (ccRCC), which accounts for 75–80% of cases. Other subtypes of renal cell carcinoma (RCC) include papillary RCC (10–15%) and chromophobe RCC (5%) [146]. The genetic and molecular mechanisms of pathogenesis and progression differ among these histologic subtypes.

#### 3.4.1. Application of NIR-PIT to Renal Cell Carcinoma

Approximately 70–75% of RCC patients are localized at initial diagnosis and are cured by surgery [147]. For small renal masses sized ≤3 cm, partial nephrectomy has been the gold standard treatment, but thermal ablation, such as cryoablation and radiofrequency ablation, are alternatives that also preserve renal function while minimizing side effects [148]. However, local recurrence is frequently observed in patients treated with thermal ablation compared with partial nephrectomy [148]. NIR-PIT may overcome these concerns as a novel nephron-sparing treatment for localized RCC. NIR light can be administered by percutaneously inserting a cylindrical light diffuser into the renal tumor under imaging guidance. Of note, NIR-PIT can spare normal nephrons due to its selectivity, thus minimizing collateral damage to normal kidney function. Thus, NIR-PIT could be a viable therapeutic option, especially for non-surgical candidates such as elderly patients with comorbidities, patients with decreased renal function, those with multiple tumors, or those with a solitary kidney.

For locally advanced RCC, neoadjuvant NIR-PIT can be combined with surgery to reduce tumor volume and extension and stimulate anti-cancer host immunity [35,36], which might lower the risk of local recurrence after surgery. Moreover, NIR-PIT may be applied to the treatment of metastatic RCC. Cytoreductive surgery, such as nephrectomy and metastasectomy, can yield a survival benefit in selected patients with metastatic RCC [149]. Thus, NIR-PIT might be utilized as a less invasive cytoreductive treatment for metastatic RCC. Moreover, a combination of NIR-PIT with ICIs may be used as a therapeutic strategy for metastatic RCC.

#### 3.4.2. Target Molecules for NIR-PIT in Renal Cell Carcinoma

To date, there have been no preclinical studies of NIR-PIT in renal cell carcinoma mouse models. Since molecular events are distinct among the histologic subtypes, candidate target molecules for NIR-PIT would be explored according to the histologic subtype. Candidate target molecules for NIR-PIT against renal cell carcinoma are described below.

##### EGFR

EGFR overexpression was observed in 84% of ccRCC, in 68% of papillary RCC, and in 75% of chromophobe RCC [150]. Thus, EGFR-targeted NIR-PIT may be effective in the majority of RCC cases.

##### PD-L1

PD-L1 is highly expressed in 24% of ccRCC compared with 11% in non-ccRCC [151]. Given that PD-L1 blockade by avelumab in combination with axitinib, a vascular endothelial growth factor receptor (VEGFR) inhibitor, is effective as a first-line treatment for advanced ccRCC [152], PD-L1 may be a potential target molecule for NIR-PIT, especially in ccRCC.

##### CTLA4 or CD25

The presence of tumor-infiltrating Tregs is associated with poor survival in ccRCC [153,154]. Thus, Tregs may be a potential target for NIR-PIT in ccRCC. Given that ipilimumab, a humanized IgG1 monoclonal antibody against CTLA-4, is already used as a standard of care in ccRCC [155], CTLA-4-targeted NIR-PIT may find a role in the treatment of ccRCC.

##### VEGFR-2

ccRCC is a highly vascularized tumor. Tyrosine kinase inhibitors targeting VEGFR, such as sorafenib, sunitinib, axitinib, and cabozantinib, show anti-cancer efficacy against ccRCC. Cancer neovasculature-targeted NIR-PIT, targeting VEGFR-2, for instance, showed anti-cancer effects in xenograft mouse models of gastric cancer [156] Cancer neovasculature-targeted NIR-PIT is a viable strategy for treating ccRCC.

### 3.5. Testicular Cancer

Testicular cancer is relatively rare, but it is the most commonly diagnosed malignancy in males aged 20 to 39 years [157]. Germ cell tumors account for 95% of testicular cancer and they are categorized into seminomas and non-seminomas [158]. Non-seminomas, which include embryonal carcinoma, choriocarcinoma, yolk sac tumors, and teratoma, are clinically and biologically aggressive, whereas seminomas usually have an indolent clinical course [158]. Patients suspected of testicular cancer undergo high inguinal orchiectomy for diagnostic and therapeutic purposes. After the pathological diagnosis of germ cell tumors, therapeutic strategies are determined based on tumor stage and whether it is seminomatous or non-seminomatous [158].

#### 3.5.1. Application of NIR-PIT to Testicular Cancer

Since germ cell tumors are extremely chemosensitive, chemotherapy is a mainstay of treatment for germ cell tumors, especially in patients with metastatic testicular cancer [159]. Thus, NIR-PIT might be utilized to treat chemorefractory metastatic lesions, though it depends on the site of the lesions.

#### 3.5.2. Target Molecules for NIR-PIT in Testicular Cancer

The efficacy of NIR-PIT for germ cell tumors has never been studied scientifically. Since seminomas and non-seminomas are molecularly distinct, candidate target molecules for NIR-PIT can be different as well. Podoplanin (PDPN) is a type I transmembrane glycoprotein expressed in various normal cells, such as type I lung alveolar cells, kidney podocytes, and lymphatic endothelial cells [160]. A previous study showed that PDPN is overexpressed in all seminoma cases. Thus, PDPN can be a potential target molecule for NIR-PIT in the treatment of seminomas. The efficacy of PDPN-targeted NIR-PIT was already reported in malignant pleural mesothelioma [161]. In a potential scenario, residual seminomatous retroperitoneal masses could be treated with laparoscopic NIR-PIT targeting PDPN.

A previous study reported that 43% of chemorefractory embryonal carcinomas expressed EGFR [162]. Thus, EGFR may be a good target for NIR-PIT in chemorefractory embryonal carcinoma.

c-KIT is a type III receptor tyrosine kinase that plays a crucial role in hematopoiesis, pigmentation, and spermatogenesis [163]. Moreover, c-KIT is a classic proto-oncogene that is involved in the uncontrolled proliferation of cancer cells [164]. c-KIT was reported to be overexpressed in 48% of chemorefractory non-seminomatous germ cell tumors [165]. Thus, c-KIT-targeted NIR-PIT, whose efficacy was investigated in in vivo gastrointestinal stromal tumor mouse models [166], may be utilized to treat chemorefractory non-seminomatous germ cell tumors.

### 3.6. Penile Cancer

Penile cancer is rare cancer that accounts for less than 1% of all malignancies worldwide [62]. Histologically, more than 95% of penile cancers are squamous cell carcinomas. Human papillomavirus infection was observed in approximately 40% of penile cancer [167]. The standard of care for penile cancer is surgical resection, and, for cases with lymph node metastasis or distant metastasis, systemic chemotherapy is required [168]. Penile cancer would be a good candidate for NIR-PIT because it is usually superficial, and non-surgical approaches would be preferred by patients. Penile cancer highly expresses epidermal growth factor receptor (EGFR). Previous studies that reported more than 90% of penile cancer showed high EGFR expression [169]. Thus, EGFR would be a good target molecule for NIR-PIT in penile cancer.

## 4. Conclusions

Although currently most clinical data has come from patients with HNSCC, NIR-PIT holds great potential as a viable treatment for a range of urologic cancers. NIR-PIT has the advantage of being highly selective, and thus causes no or minimal damage to healthy tissues in the surrounding area. This is of particular importance in urologic cancers where minimal collateral damage equates to better functional status after treatment. Thus, NIR-PIT could achieve organ and function preservation while it eradicates cancer cells. Moreover, NIR-PIT substantially activates the anti-cancer host immunity through the induction of ICD. The combination of NIR-PIT with immune activation therapies such as ICIs is a promising strategy for treating urologic cancers. In the near future, NIR-PIT may create a paradigm shift in the clinical practice for urologic cancers.

## Figures and Tables

**Figure 1 cancers-14-02996-f001:**
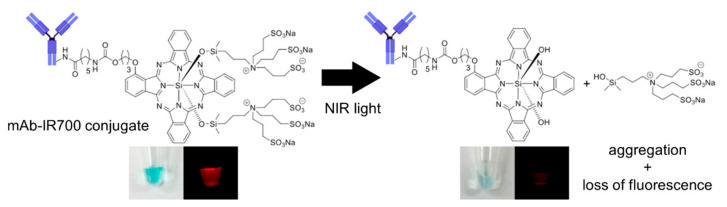
Scheme of IR700 chemical structure and its conformational change upon NIR light irradiation. Adapted with permission from Ref. [7]. Copyright 2018 American Chemical Society.

**Figure 2 cancers-14-02996-f002:**
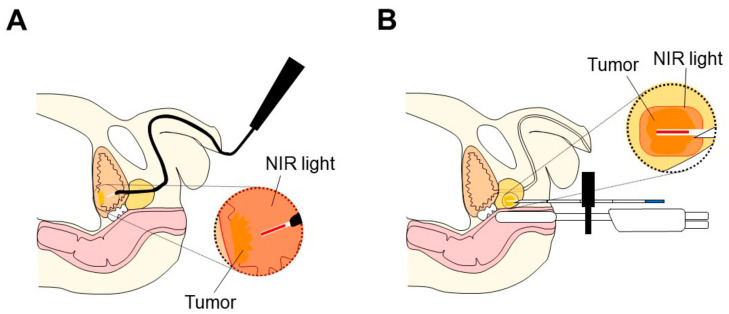
Scheme for NIR light irradiation to treat bladder and prostate cancer. NIR light is irradiated to a bladder tumor using a cylindrical diffuser via a cystoscope (**A**). NIR light is irradiated to a prostate tumor using a cylindrical diffuser via a needle (**B**).

**Figure 3 cancers-14-02996-f003:**
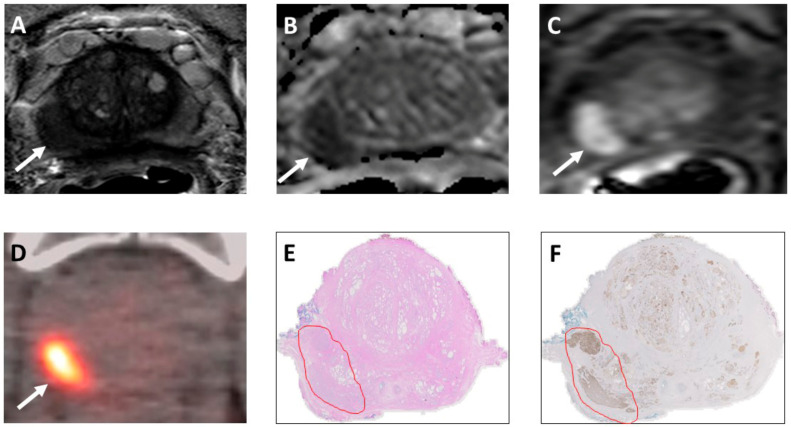
71-year-old male patient with a serum PSA of 5.2 ng/mL. Axial T2-weighted (T2W) MRI shows a hypointense lesion in the right base peripheral zone (arrow) (**A**), which shows diffusion restriction on apparent diffusion coefficient (ADC) map (arrow) (**B**) and early focal enhancement on dynamic contrast enhanced (DCE) MRI (arrow) (**C**). 18-F-DCFPyL PET/CT shows focal uptake within the right-sided lesion (arrow) (**D**). Hematoxylin-eosin (HE) histopathology slide confirms presence of Gleason 4 + 4 prostate cancer within this lesion (inked in red) (**E**). PSMA immunohistochemistry (IHC) staining shows selective PSMA expression within the right base peripheral zone lesion (inked in red) (**F**).

**Table 1 cancers-14-02996-t001:** Candidate target molecules for urologic cancers in cancer cell-targeted NIR-PIT.

Type	Target Molecules												
	EGFR	HER2	PSMA	CD44	PDPN	CD47	TROP2	PD-L1	c-KIT	GPR87	GPC1	Nectin4	FGFR3
Bladder cancer	++	+		+	±	+	+	+		+	+	++	+
Prostate cancer	±		++	+		+	+	+			+		
Renal cell carcinoma	+	±		±				+					
Upper tract urothelial cancer	+	+		+			+	+					
Testicular cancer	±	±		+	+	+		+	+				
Penile cancer	++			+	+			+					

EGFR, epidermal growth factor receptor; HER2, human epidermal growth factor receptor-2; PSMA, prostate-specific membrane antigen; PDPN, podoplanin; TROP-2, tumor-associated calcium signal transducer 2; PD-L1, programmed death-ligand 1; GPR87, G-protein-coupled receptor 87; GPC1, Glypican-1; FGFR3, fibroblast growth factor receptor 3; ++, high expression; +, moderate expression; ±, weak expression. In most preclinical studies against urologic cancer, APCs were injected once (day 0) and NIR light was irradicated twice (day 1: 50 J/cm^2^, day 2: 100 J/cm^2^). Data from Ref. [61].

## Data Availability

The data presented in this study are available in the present paper.

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
