# Peer review of "Near-Infrared Photoimmunotherapy (NIR-PIT) in Urologic Cancers"

_cancers, 2022, doi:10.3390/cancers14122996_

Round 1
Reviewer 1 Report
In the review article by Fukushima et al. the authors describe the potential utility of PIT in urologic cancers. While the review article is interesting and important given the clinical success of PIT there are a few suggestions that may help improve the manuscript.
General comments:
1. The authors use the term “NIR-PIT” for the APC developed in the present study and state in lines 70-72 that “This cytotoxic mechanism distinguishes NIR-PIT from conventional photodynamic therapy (PDT) and photothermal therapy (PTT), which require the generation of reactive oxygen species to cause non-specific damage to adjacent normal tissues.” While it is agreed that the mechanism of PIT using IRDye700 differs from that of other photosensitizers that function mainly through ROS generation, the term PIT was initially referred for antibody photosensitizer conjugates that function through ROS generation (Mew et al Journal of Immunology, 1983). The authors are suggested to acknowledge that the phenomenon of PIT is also used for antibody-photosensitizer conjugates and clarify that this review is focused specifically on the PIT mediated by IRDye700.
2. In section 2.2 the authors make a statement “Most other cancer therapies cause apoptotic cell death which is more organized or “programmed” and less immunogenic.”. The authors should provide a reference for this. Immunogenic apoptosis is a well-established phenomenon, and if there is a comparison suggesting that necrosis is more immunogenic than apoptosis that should be cited and the statement should be modified.
3. The section “3.1.2. Target molecules for NIR-PIT in bladder cancer” lists the potential molecular targets for PIT, however the section also includes “Tregs” as a separate heading. For maintaining consistency, the section should have molecular targets CTLA4- or CD25- as headings.
4. The heading in line 282 “Selection of target molecules for NIR-PIT according to molecular subtypes” should be in the beginning of the section 3.1.2., rather than at the end where it is now placed.
5. Similarly in section 3.4.2 the subheadings should just be molecular targets and not cells – Tregs and neovasculature.
Author Response
Comments and Suggestions for Authors
In the review article by Fukushima et al. the authors describe the potential utility of PIT in urologic cancers. While the review article is interesting and important given the clinical success of PIT there are a few suggestions that may help improve the manuscript.
We greatly appreciated you for taking time to review our paper and giving instructive comments. We revised the manuscript according to your comments. Please check the changes we made in the revised version outlined below.
General comments:
- The authors use the term “NIR-PIT” for the APC developed in the present study and state in lines 70-72 that “This cytotoxic mechanism distinguishes NIR-PIT from conventional photodynamic therapy (PDT) and photothermal therapy (PTT), which require the generation of reactive oxygen species to cause non-specific damage to adjacent normal tissues.” While it is agreed that the mechanism of PIT using IRDye700 differs from that of other photosensitizers that function mainly through ROS generation, the term PIT was initially referred for antibody photosensitizer conjugates that function through ROS generation (Mew et al Journal of Immunology, 1983). The authors are suggested to acknowledge that the phenomenon of PIT is also used for antibody-photosensitizer conjugates and clarify that this review is focused specifically on the PIT mediated by IRDye700.
As you mentioned, the term ‘photoimmunotherapy’ was once used to mean targeted photodynamic therapy using antibodies conjugated with conventional photosensitizers such as hematoporphyrin based on cytotoxicity induced by ROS in 1983. However, NIR-PIT is totally different from this ‘old photoimmunotherapy’.
In ‘old photoimmunotherapy, its cytotoxicity depends on ROS. Thus, it causes significant non-specific damage to normal cells. In contrast, NIR-PIT showed selective cytotoxicity on target cells based on a photo-induced ligand release reaction rather than ROS after systemic intravenous injection of IR700-based APCs.
Moreover, in ‘old photoimmunotherapy’, therapeutic effects were shown only in vitro, or in tumors with intra-tumoral or intra-spatial administration in vivo because of the unfavorable chemical design that conjugated antibodies with hydrophobic chemicals. Conjugates were insufficiently delivered to the tumor due to the rapid liver accumulation of conjugates promoted by the hydrophobicity of conventional photosensitizers. On the other hand, this is not an issue for NIR-PIT because IR700-based APCs are highly hydrophilic. Thus, NIR-PIT can be safely applied to patients and was approved for clinical use in Japan.
Therefore, we added the following sentences to explain that NIR-PIT is totally different from the ‘old photoimmunotherapy’:
”In 1983, the term “photoimmunotherapy” was used to imply a targeted photodynamic therapy using antibodies that kills target cells based on reactive oxygen species (ROS)-induced cytotoxicity using antibodies conjugated to conventional photosensitizers such as hematoporphyrin.21 However, it was not successfully used as an anti-cancer treatment via systemic administration.22 This is because the hydrophobic nature of conventional photosensitizers causes rapid liver accumulation of conjugates, leading to insufficient delivery of conjugates to the tumor.23 Moreover, because the cell killing efficacy of this “old photoimmunotherapy” depends on ROS that mostly pro-duces out of the target cells, it causes significant non-specific damage to adjacent nor-mal cells. In contrast, NIR-PIT selectively kills APC-bound target cells by the immuno-genic cell death as a result of photo-induced ligand release reactions rather than ROS reactions after intravenous injection of IR700-based APCs (Figure 1). Therefore, as a consequence, anti-cancer host immune response is rationally induced after NIR-PIT. Thus, NIR-PIT totally differs from the “old photoimmunotherapy.” (Page 2, Line 58-68)
- In section 2.2 the authors make a statement “Most other cancer therapies cause apoptotic cell death which is more organized or “programmed” and less immunogenic.”. The authors should provide a reference for this. Immunogenic apoptosis is a well-established phenomenon, and if there is a comparison suggesting that necrosis is more immunogenic than apoptosis that should be cited and the statement should be modified.
Thank you for your suggestion. We added two references (40, 41) that state chemotherapy and radiotherapy induce apoptosis. We could not find researches that directly compared immunogenicity between apoptosis and necrosis. However, several studies reported apoptosis is immunogenically silent and immunosuppressive whereas necrosis is pro-inflammatory (42, 43). This suggests that apoptosis is generally considered as less immunogenic even though recent studies have proposed immunogenic apoptosis. Thus, we revised this sentence as follows:
“Most other cancer therapies such as chemotherapy and radiotherapy cause apoptotic cell death,40, 41 which is generally more organized or “programmed” and less immunogenic.42, 43” (Page 3, Line 106-108)
- Hannun YA. Apoptosis and the dilemma of cancer chemotherapy. Blood. 1997; 89: 1845-1853.
- Balcer-Kubiczek EK. Apoptosis in radiation therapy: a double-edged sword. Exp Oncol. 2012; 34: 277-285.
- Voll RE, Herrmann M, Roth EA, Stach C, Kalden JR, Girkontaite I. Immunosuppressive effects of apoptotic cells. Nature. 1997; 390: 350-351.
- Newton K, Manning G. Necroptosis and Inflammation. Annu Rev Biochem. 2016; 85: 743-763.
- The section “3.1.2. Target molecules for NIR-PIT in bladder cancer” lists the potential molecular targets for PIT, however the section also includes “Tregs” as a separate heading. For maintaining consistency, the section should have molecular targets CTLA4- or CD25- as headings.
As you pointed out, we changed “Tregs” into “CTLA4 or CD25”.
- The heading in line 282 “Selection of target molecules for NIR-PIT according to molecular subtypes” should be in the beginning of the section 3.1.2., rather than at the end where it is now placed.
Thank you for your constructive comment. The subsection “Selection of target molecules for NIR-PIT according to molecular subtypes” discusses several target molecules including EGFR and HER2, and thus we think that it is better to discuss it after commenting on candidate target molecules for bladder cancer. Therefore, we kept this subsection as such.
- Similarly in section 3.4.2 the subheadings should just be molecular targets and not cells – Tregs and neovasculature.
We changed “Tregs” into “CTLA4 or CD25”. In addition, we changed “Cancer neovasculature” into “VEGFR-2”.

Reviewer 2 Report
The presented work is an overview of application of the IRDye700DX (IR700) dye conjugated with targeting antibodies for urologic cancer treatment, which is an extremely important task in modern medicine. This overview can be of great interest for researchers involved in the treatment of urologic cancer and therefore I suggest it to be published after minor corrections.
1) My most important concern is terminology. From one side, authors call the suggested treatment "photoimmunotherapy (PIT)" - an oncological treatment that combines photodynamic therapy of tumor with immunotherapy. From another side, authors note that the mechanism of IR700 action consists in "APC aggregation and conformational changes that damage the adjacent cell membrane" (rows 66-68) and therefore the "cytotoxic mechanism (for IR700) distinguishes NIR-PIT from conventional photodynamic therapy (PDT) ..., which require the generation of reactive oxygen species to cause non-specific damage to adjacent normal tissues" (rows 70-72). So, it is unclear, what is the real mechanism of the IR700 action, photodynamic (reactive species generation) or conformational changes-initiated damage of the cell membranes. In the second case, I am not sure that authors can use term "photoimmunotherapy (PIT)" in the title and all over the text. In both cases, term antibody-photoabsorber conjugate (APC) sounds not clear because the "job" of this conjugate is not only to absorb light but also either generate reactive species or initiate damage to cell membranes by conformational changes.
2) I suggest authors to show the IRDye700DX (IR700) chemical structure and scheme of the supposed/confirmed transformation upon light irradiation (row 30).
3) Starting from the Section 2, authors do not mention anymore any IR700 conjugates and therefore it is not clear, if the following discussion relates only to this dye and its conjugates or to some other dyes/conjugates. Authors should mention clearly, which dye/APC is used in the cited publications starting from the Section 2. Is it the same IR700 or something different?
4) "the tumor is exposed to relatively low energy 690 nm NIR light" (rows 63-64). Energy is proportional to 1/wavelength and therefore it is not clear what means this statement. Maybe authors mean light dose (which has nothing to do with energy)? By the way, this is a shortcoming of the presented review: light dose is not mention at all and therefore it is not clear how efficient is the IR700 dye (photosensitizer?) and it is not possible to compare its efficacy with many other available and approved photosensitizers.
5) " Most other cancer therapies cause apoptotic cell death" (row 93). Please indicate these therapies and provide references.
6) Term "the tumor microenvironment" is not clear in this context. Please explain.
7) "nano-sized compound" (row 133). This term is unclear. Please provide examples of these nano-sized compounds with appropriate citations. Proteins, dendrimers, polymers, something else?
8) Table 1:
(1) I would suggest adding citations in the table.
(2) Does it possible to indicate light dose applied for each treatment? Otherwise, please note a range of light doses used for urologic cancer treatment.
(3) What is a general treatment procedure? How the treatment is performed? How many times APC is injected and how many times light-irradiated + light doses?
9) Correct increased fonts.

Author Response
Comments and Suggestions for Authors
The presented work is an overview of application of the IRDye700DX (IR700) dye conjugated with targeting antibodies for urologic cancer treatment, which is an extremely important task in modern medicine. This overview can be of great interest for researchers involved in the treatment of urologic cancer and therefore I suggest it to be published after minor corrections.
We greatly appreciated you for taking time to review our paper and giving instructive comments. We revised the manuscript according to your comments. Please check the changes we made in the revised version outlined below.
1) My most important concern is terminology. From one side, authors call the suggested treatment "photoimmunotherapy (PIT)" - an oncological treatment that combines photodynamic therapy of tumor with immunotherapy. From another side, authors note that the mechanism of IR700 action consists in "APC aggregation and conformational changes that damage the adjacent cell membrane" (rows 66-68) and therefore the "cytotoxic mechanism (for IR700) distinguishes NIR-PIT from conventional photodynamic therapy (PDT) ..., which require the generation of reactive oxygen species to cause non-specific damage to adjacent normal tissues" (rows 70-72). So, it is unclear, what is the real mechanism of the IR700 action, photodynamic (reactive species generation) or conformational changes-initiated damage of the cell membranes. In the second case, I am not sure that authors can use term "photoimmunotherapy (PIT)" in the title and all over the text. In both cases, term antibody-photoabsorber conjugate (APC) sounds not clear because the "job" of this conjugate is not only to absorb light but also either generate reactive species or initiate damage to cell membranes by conformational changes.
Thank you for your pointing out. Historically, the term ‘photoimmunotherapy’ was once used to mean targeted photodynamic therapy using antibodies conjugated with conventional photosensitizers such as hematoporphyrin based on cytotoxicity induced by ROS in 1983. However, NIR-PIT is totally different from this ‘old photoimmunotherapy’.
In ‘old photoimmunotherapy, its cytotoxicity depends on ROS. Thus, it causes significant non-specific damage to normal cells. In contrast, NIR-PIT showed selective cytotoxicity on target cells based on a photo-induced ligand release reaction rather than ROS after systemic intravenous injection of IR700-based APCs.
Moreover, in ‘old photoimmunotherapy’, therapeutic effects were shown only in vitro, or in tumors with intra-tumoral or intra-spatial administration in vivo because of the unfavorable chemical design that conjugated antibodies with hydrophobic chemicals. Conjugates were insufficiently delivered to the tumor due to the rapid liver accumulation of conjugates promoted by the hydrophobicity of conventional photosensitizers. On the other hand, this is not an issue for NIR-PIT because IR700-based APCs are highly hydrophilic. Thus, NIR-PIT can be safely applied to patients and was approved for clinical use in Japan.
In the present day, the term ‘NIR-PIT’ is widely recognized as ‘the cancer therapy that utilizes antibody-IR700 complex and induces cell membrane destruction leading to necrotic cell death upon the irradiation of NIR light.’
The term ‘APC’ simply means ‘antibodies conjugated to photoabsorber IR700. The term ‘APC’ in itself does not imply that it produces ROS. Thus, we used the term ‘APC’ in this article.
Therefore, we added the following sentences to explain that NIR-PIT is totally different from the ‘old photoimmunotherapy’:
”In 1983, the term “photoimmunotherapy” was used to imply a targeted photodynamic therapy using antibodies that kills target cells based on reactive oxygen species (ROS)-induced cytotoxicity using antibodies conjugated to conventional photosensitizers such as hematoporphyrin.21 However, it was not successfully used as an anti-cancer treatment via systemic administration.22 This is because the hydrophobic nature of conventional photosensitizers causes rapid liver accumulation of conjugates, leading to insufficient delivery of conjugates to the tumor.23 Moreover, because the cell killing efficacy of this “old photoimmunotherapy” depends on ROS that mostly pro-duces out of the target cells, it causes significant non-specific damage to adjacent nor-mal cells. In contrast, NIR-PIT selectively kills APC-bound target cells by the immuno-genic cell death as a result of photo-induced ligand release reactions rather than ROS reactions after intravenous injection of IR700-based APCs (Figure 1). Therefore, as a consequence, anti-cancer host immune response is rationally induced after NIR-PIT. Thus, NIR-PIT totally differs from the “old photoimmunotherapy.” (Page 2, Line 58-68)
2) I suggest authors to show the IRDye700DX (IR700) chemical structure and scheme of the supposed/confirmed transformation upon light irradiation (row 30).
Thank you for your constructive comment. I added the scheme of IR700 chemical structure and its transformation upon NIR light irradiation (Figure 1).
3) Starting from the Section 2, authors do not mention anymore any IR700 conjugates and therefore it is not clear, if the following discussion relates only to this dye and its conjugates or to some other dyes/conjugates. Authors should mention clearly, which dye/APC is used in the cited publications starting from the Section 2. Is it the same IR700 or something different?
Thank you for your suggestion. All the cited publications starting from Section 2 used IR700. In NIR-PIT, only IR700 is used to construct APCs. For clarity, we used the term “IR700-based APC” in several parts of the Section 2 and revised several sentences as follows:
“In contrast, NIR-PIT showed selective cytotoxicity on target cells based on a photo-induced ligand release reaction rather than ROS after systemic intravenous injection of IR700-based APCs.” (Page 2, Line 65-68)
“NIR-PIT employs IR700-based APCs, antibodies conjugated to the IR700 silica-phthalocyanine dye, which absorb NIR photons. When injected intravenously, IR700-based APCs bind to a specific cell membrane antigen on cancer cell surface. This alone is insufficient for a treatment effect.” (Page 2, Line 74-77)
4) "the tumor is exposed to relatively low energy 690 nm NIR light" (rows 63-64). Energy is proportional to 1/wavelength and therefore it is not clear what means this statement. Maybe authors mean light dose (which has nothing to do with energy)? By the way, this is a shortcoming of the presented review: light dose is not mention at all and therefore it is not clear how efficient is the IR700 dye (photosensitizer?) and it is not possible to compare its efficacy with many other available and approved photosensitizers.
As you are pointing out, the phrase “relatively low energy” was misleading. Thus, we deleted “relatively low energy” from this sentence. As for the light doses, most preclinical studies used 30-100 J/cm2 of NIR light. Thus, we revised the sentence as follows:
“However, approximately 24 hours after injection, the tumor is exposed to 690 nm NIR light from a laser usually in doses 30–100 J/cm2.” (Page2, Line 77-79)
5) " Most other cancer therapies cause apoptotic cell death" (row 93). Please indicate these therapies and provide references.
Thank you for your pointing out. We added two references (40, 41) that state chemotherapy and radiotherapy induce apoptosis. We revised this sentence as follows:
“Most other cancer therapies such as chemotherapy and radiotherapy cause apoptotic cell death,40, 41 which is generally more organized or “programmed” and less immunogenic.” (Page3, Line 106-108)
- Hannun YA. Apoptosis and the dilemma of cancer chemotherapy. Blood. 1997; 89: 1845-1853.
- Balcer-Kubiczek EK. Apoptosis in radiation therapy: a double-edged sword. Exp Oncol. 2012; 34: 277-285.
6) Term "the tumor microenvironment" is not clear in this context. Please explain.
Various cells such as cancer cells, immune cells, and cancer-associated fibroblasts are included in the tumor microenvironment. Thus, we added an explanation about “cells in the tumor microenvironment” as follows:
“In addition to targeting cancer cells, NIR-PIT can selectively target cells in the tumor microenvironment including immune cells and cancer-associated fibroblasts.” (Page 3, Line 124-125)
7) "nano-sized compound" (row 133). This term is unclear. Please provide examples of these nano-sized compounds with appropriate citations. Proteins, dendrimers, polymers, something else?
As you are pointing out, we added examples of nano-sized compounds as follows:
“For example, when NIR-PIT is combined with nano-sized agents such as liposome-containing daunorubicine or albumin-bound paclitaxel, therapeutic effects were significantly augmented compared with either therapy alone.” (Page 4, Line 151-154)
8) Table 1:
(1) I would suggest adding citations in the table.
We added a reference (61) to Table 1.
- The Human Protein Atlas. Available online: https://www.proteinatlas.org/ (accessed on March 22, 2022).
(2) Does it possible to indicate light dose applied for each treatment? Otherwise, please note a range of light doses used for urologic cancer treatment.
Thank you for your suggestion. Table 1 includes some target molecules that have never been evaluated in preclinical studies. In EGFR-, HER2-, CD47-, and PSMA-targeted NIR-PIT, 50 or 100 J/cm2 was used in preclinical studies for urologic cancer treatment. Thus, we added an explanation of light dose in Table 1 as follows;
“In most preclinical studies against urologic cancer, APCs were injected once (day 0) and NIR light was irradicated twice (day 1: 50 J/cm2, day 2: 100 J/cm2).” (Page 4, Line 170-171)
(3) What is a general treatment procedure? How the treatment is performed? How many times APC is injected and how many times light-irradiated + light doses?
In most preclinical studies against urologic cancer, APCs were injected once (day 0) and NIR light was irradicated twice (day1: 50 J/cm2, day2: 100 J/cm2). Thus, we added an explanation of treatment procedures in Table 1 as follows:
“In most preclinical studies against urologic cancer, APCs were injected once (day 0) and NIR light was irradicated twice (day 1: 50 J/cm2, day 2: 100 J/cm2).” (Page 4, Line 170-171)
9) Correct increased fonts.
Thank you for your pointing out. I corrected it.
